# Prompt treatment of fever and its associated factors among under-five children in sub-Saharan Africa: A multilevel analysis of evidence from 36 countries

**Enyew Getaneh Mekonen**[1]*, **Belayneh Shetie Workneh**[2], **Tadesse Tarik Tamir**[3], **Alebachew Ferede Zegeye**[4]

**1** Department of Surgical Nursing, School of Nursing, College of Medicine and Health Sciences, University of Gondar, Gondar, Ethiopia, **2** Department of Emergency and Critical Care Nursing, School of Nursing, College of Medicine and Health Sciences, University of Gondar, Gondar, Ethiopia, **3** Department of Pediatrics and Child Health Nursing, School of Nursing, College of Medicine and Health Sciences, University of Gondar, Gondar, Ethiopia, **4** Department of Medical Nursing, School of Nursing, College of Medicine and Health Sciences, University of Gondar, Gondar, Ethiopia

* enyewgetaneh12@gmail.com

**Data Availability Statement:** The data from 36 SSA countries is publicly available online at https://dhsprogram.com/data/available-datasets.cfm.

## Abstract

### Introduction

Despite the decrease in the global under-five mortality rate, the highest rates of mortality are reported in sub-Saharan Africa. More than one-third of all deaths among under-five children are either from lower respiratory tract infections, diarrhea, or malaria. Poor treatment-seeking behavior for fever among mothers of under-five children is a big concern in sub-Saharan Africa. However, the pooled prevalence of prompt treatment of fever and its associated factors among under-five children in the region using nationally representative data is not known. Therefore, the findings of this study will inform policymakers and program managers who work on child health to design interventions to improve the timely and appropriate treatment of fever among under-five children.

### Methods

Data from the recent demographic and health surveys of 36 countries in sub-Saharan Africa conducted between 2006 and 2022 were used. A total weighted sample of 71,503 living children aged under five years with a fever was included in the study. Data extracted from DHS data sets were cleaned, recorded, and analyzed using STATA/SE version 14.0 statistical software. Multilevel mixed-effects logistic regression was used to determine the factors associated with the outcome variable. Intra-class correlation coefficient, likelihood ratio test, median odds ratio, and deviance (-2LLR) values were used for model comparison and fitness. Finally, variables with a p-value <0.05 and an adjusted odds ratio with a 95% confidence interval were declared statistically significant.

**Funding:** The author(s) received no specific funding for this work.

**Abbreviations:** ANC, Antenatal Care; AOR, Adjusted Odds Ratio; CI, Confidence Interval; DHS, Demographic and Health Survey; ICC, Intra-class Correlation Coefficient; LMICs, low- and middle-income countries; MOR, Median Odds Ratio; PCV, Proportional Change in Variance; SSA, sub-Saharan Africa; VIF, Variance Inflation Factor; WHO, World Health Organization.

## Results

The pooled prevalence of prompt treatment of fever among under-five children in sub-Saharan African countries was 26.11% (95% CI: 25.79%, 26.44%). Factors like maternal education [AOR = 1.18; 95% CI (1.13, 1.25)], maternal working status [AOR = 1.34; 95% CI (1.27, 1.41)], media exposure [AOR = 1.05; 95% CI (1.01, 1.10)], household wealth index [AOR = 1.13; 95% CI (1.06, 1.19)], distance to a health facility [AOR = 1.18; 95% CI (1.13, 1.23)], healthcare decisions [AOR = 1.34; 95% CI (1.01, 1.77)], visited healthcare facility last 12 months [AOR = 1.45; 95% CI (1.38, 1.52)], antenatal care attendance [AOR = 1.79; 95% CI (1.61, 1.99)], place of delivery [AOR = 1.55; 95% CI (1.47, 1.63)], and community-level antenatal care utilization [AOR = 1.08; 95% CI (1.02,1.14)] were significantly associated with prompt treatment of fever among under-five children.

## Conclusion

The pooled prevalence of prompt treatment of fever among under-five children in sub-Saharan African countries was low. Educated women, working mothers, having media exposure, rich household wealth status, perceiving distance to a health facility was not a big problem, making healthcare decisions with husband or partner, visiting healthcare facility in the last 12 months, antenatal care attendance, health facility delivery, and high community-level antenatal care utilization increase the odds of prompt treatment of fever. Therefore, women's empowerment, information dissemination through mass media, maintaining regular visits to healthcare facilities, and strengthening health facility delivery and antenatal care services are strongly recommended.

## Introduction

Preventable and treatable causes are responsible for the deaths of an estimated five million under-five children [1]. Despite the decrease in the global under-five mortality rate, the highest rates of mortality are reported in sub-Saharan Africa (SSA), which is 14 times higher than Europe and North America [1]. Mortality and morbidity secondary to infection in under-five children is one of the most challenging health problems to address, as signs and symptoms are often broad and illnesses can quickly progress to severe disease [2]. More than one-third of all deaths among under-five children in low- and middle-income countries (LMICs) are either from lower respiratory tract infections, diarrhea, or malaria [3]. These deaths could be prevented by quick diagnosis and treatment, including antibiotics for lower respiratory infections (pneumonia), oral rehydration therapy for diarrhea, and antimalarial drugs for malaria [4].

Fever can be defined as an elevation of body temperature above 37.2°C (axillary), 37.5°C (oral), or 38°C (rectal and tympanic) [5]. It is a significant public health problem among children aged 0 to 59 months in SSA, with a prevalence of 22.65% [6]. Fever is the most common symptom associated with most childhood diseases and the primary cause of pediatric health facility consultations and hospitalizations in under-five children [7, 8]. In SSA, the major causes of fevers are upper and lower respiratory infections and probably viral infections, which cannot be treated using antibiotics or antimalarial drugs [6]. Children from LMICs encounter around 40 or more episodes of fever by the age of five, which healthcare professionals tend to manage with antimalarial drugs or antibiotics [9].

A child presented with a complaint of fever might be diagnosed with one of the common childhood febrile illnesses, including sepsis, meningitis, herpes simplex virus, urinary tract infections, group B streptococcus disease, acute respiratory infections, enteric fever, or malaria [10, 11]. Despite being the largest killer of these diseases among children under the age of five, care seeking for febrile children often occurs too late, making community-based efforts vital to increasing access to prompt treatment [12]. Delayed treatment of fever cases among under-five children was 31.1% in India [13], 55.4% in Tanzania [14], and 62.1% in Nigeria [15]. It is well known that fever has some potentially harmful effects if it is not managed timely and appropriately [16]. Inhibited gastric emptying and impaired intestinal absorption, with accompanying anorexia and sometimes vomiting secondary to the sympathetic nervous system response to fever, are common in febrile children [17].

Studies conducted elsewhere showed that the prevalence of prompt treatment of fever was 68.9% in India [13], 37.9% in Nigeria [15], 44.6% in Tanzania [14], 42% in Mazabuka District, Zambia [18], and 57.3% in Zambia [19]. Factors like child age [15], household wealth status [13, 15], mother's occupation [15], antenatal care (ANC) visits during pregnancy [15], pregnancy intention [15], number of under-five children in the household [14], distance to health-care facility [13, 14], residence [13], child sex [19], and mother's education [19] were significantly associated with treatment of fever among under-five children.

The World Health Organization (WHO) recommends increasing efforts to achieve Sustainable Development Goal 3.2. 1 (i.e., to end preventable deaths of under-five children by 2030 and reduce under-five mortality to at least as low as 25 per 1000 live births in every country) in SSA [1]. However, poor treatment-seeking behavior for fever among mothers of under-five children is a big concern in the region and has a vast contribution to the morbidity and mortality of under-five children [14, 20]. As far as our knowledge is concerned, the pooled prevalence of prompt treatment of fever and its associated factors among under-five children in SSA using nationally representative data is not known. Therefore, the findings of this study will inform policymakers and governmental and non-governmental organizations that work on child health to design interventions to improve the timely and appropriate treatment of fever among under-five children.

## Materials and methods

### Data sources, study design, and sampling

A cross-sectional pooled dataset using the recent DHS data from 36 SSA countries, which was conducted between 2006 and 2022, was employed. Demographic and health surveys from 36 SSA countries, including Angola (2015–16), Burkina Faso (2010), Benin (2017–18), Burundi (2016–17), Congo Democratic Republic (2013–14), Congo (2011–12), Cote d'Ivoire (2011–12), Cameroon (2022), Ethiopia (2016), Gabon (2019–21), Ghana (2014), Gambia (2019–20), Guinea (2018), Kenya (2022), Comoros (2012), Liberia (2019–20), Lesotho (2014), Madagascar (2008), Mali (2018), Malawi (2015–16), Mozambique (2011), Nigeria (2018), Niger (2019), Namibia (2013), Rwanda (2019–20), Sierra Leone (2019), Senegal (2019), Sao Tome and Principe (2008–09), Swaziland (2006–07), Chad (2014–15), Togo (2013–14), and Tanzania (2022), Uganda (2016), South Africa (2016), Zambia (2018), Zimbabwe (2015), were used. The data were appended to figure out the pooled prevalence of prompt treatment of fever and its associated factors in SSA countries. Different datasets, including those for children, males, women, births, and households, are included in the survey for each country. For this study, the kid's record (KR) file was used. The DHS is a nationwide survey, mostly collected every five years across LMICs. It makes cross-country comparison possible as it uses standard procedures for sampling, questionnaires, data collection, cleaning, coding, and analysis [21]. A total weighted

sample of 71,503 living children aged under five years with a fever was included in the current study (Table 1). The DHS employs a stratified, two-stage sampling technique [22]. The first stage involves the development of a sampling frame, consisting of a list of primary sampling units (PSUs) or enumeration areas (EAs), which covers the entire country and is usually developed from the latest available national census. The second stage is the systematic sampling of households listed in each cluster, or EA. Further information on the survey sampling strategies is available in the DHS guideline [23].

**Table 1. Sample size for prevalence and associated factors of prompt treatment of fever among under-five children in sub-Saharan African countries.**

| Country | Year of survey | Weighted sample (n) | Weighted sample (%) |
|---|---|---|---|
| Angola | 2015–16 | 1,941 | 2.71 |
| Burkina Faso | 2010 | 2,958 | 4.14 |
| Benin | 2017–18 | 2,398 | 3.35 |
| Burundi | 2016–17 | 4,528 | 6.33 |
| Congo Democratic Republic | 2013–14 | 5,157 | 7.21 |
| Congo | 2011–12 | 2,297 | 3.21 |
| Cote d'Ivoire | 2011–12 | 1,657 | 2.32 |
| Cameroon | 2022 | 1,330 | 1.86 |
| Ethiopia | 2016 | 1,350 | 1.89 |
| Gabon | 2019–21 | 1,309 | 1.83 |
| Ghana | 2014 | 822 | 1.15 |
| Gambia | 2019–20 | 1,315 | 1.84 |
| Guinea | 2018 | 1,221 | 1.71 |
| Kenya | 2022 | 3,145 | 4.4 |
| Comoros | 2012 | 635 | 0.89 |
| Liberia | 2019–20 | 1,471 | 2.06 |
| Lesotho | 2014 | 390 | 0.55 |
| Madagascar | 2008 | 1,124 | 1.57 |
| Mali | 2018 | 1,470 | 2.06 |
| Malawi | 2015–16 | 4,690 | 6.56 |
| Mozambique | 2011 | 1,313 | 1.84 |
| Nigeria | 2018 | 7,518 | 10.51 |
| Niger | 2019 | 1,563 | 2.19 |
| Namibia | 2013 | 1,138 | 1.59 |
| Rwanda | 2019–20 | 1,467 | 2.05 |
| Sierra Leone | 2019 | 1,469 | 2.05 |
| Senegal | 2019 | 922 | 1.29 |
| Sao Tome and Principe | 2008–09 | 249 | 0.35 |
| Swaziland | 2006–07 | 712 | 1.00 |
| Chad | 2014–15 | 3,539 | 4.95 |
| Togo | 2013–14 | 1,413 | 1.98 |
| Tanzania | 2022 | 1,012 | 1.42 |
| Uganda | 2016 | 4,987 | 6.97 |
| South Africa | 2016 | 647 | 0.90 |
| Zambia | 2018 | 1,549 | 2.17 |
| Zimbabwe | 2015 | 797 | 1.11 |
| Total sample size | | 71,503 | 100 |

## Variables of the study

**Dependent variable.** The outcome variable for this study was prompt treatment of fever (yes = 1, no = 0). Prompt treatment of fever was defined as treatment sought the same or next day following the onset of fever in under-five children as reported by mothers in the last two weeks preceding the interview [23].

**Independent variables.** The current study considered both individual and community-level variables. Individual level variables: mother's age (15–24 years, 25–34 years, 35–49 years), mother's education (no formal education, primary education, secondary or higher education), mother's current marital status (unmarried, married), mother's working status (not working, working), media exposure (no, yes), household wealth index (poor, middle, rich), distance to the health facility (big problem, not a big problem), health care decisions (mothers alone, with husband/partner, husband/partner alone, others), visited health facility last 12 months (no, yes), pregnancy intention (unintended, intended), frequency of antenatal care use (none, 1–3 visits, 4+ visits), place of delivery (home, health facility), child age (0–11 months, 12–23 months, 24–59 months), child sex (male, female), and preceding birth interval (<24 months, ≥24 months). Community-level variables: place of residence (urban, rural), community-level media exposure (low, high), community-level education (low, high), community poverty level (low, high), community-level ANC utilization (low, high), and community-level health facility delivery (low, high). These factors were created by aggregating individual-level factors, as these factors were not directly available from DHS data.

## Description of independent variables

**Media exposure.** Created by combining whether a respondent reads newspapers or magazines, listens to the radio, or watches television, and is coded as "yes" if the mother was exposed to at least one of these media and "no" otherwise.

**Pregnancy intention.** Re-categorized as intended (if the pregnancy was wanted) and unintended (incorporating both mistimed and unintended).

**Community-level of media exposure.** The proportion of women who had been exposed to at least one media (television, radio, or newspaper) and categorized based on the national median value as low (communities with ≤50% of women exposed) and high (communities with >50% of women exposed).

**Community-level education.** The proportion of women with a minimum primary level of education derived from data on respondents' level of education. Then, it was categorized using the national median value into two categories: low (communities with ≤50% of women having at least primary education) and high (communities with >50% of women having at least primary education).

**Community-level health facility delivery.** The proportion of women with health facility delivery and recoded as low (communities with ≤50% of women delivered at the health facility) and high (communities with >50% of women have delivered at the health facility) community level of health facility delivery.

**Community-level ANC utilization.** The proportion of women with a minimum of four or more ANC visits. It is categorized using the national median value as follows: low (communities with ≤50% of women have at least four ANC visits) and high (communities with >50% of women have at least four ANC visits).

**Community poverty level.** An aggregated variable from household wealth status (proportion of women from poor and rich wealth status), and it was recoded as low and high community poverty level, as above.

## Data management and analysis

Data extracted from the recent DHS data sets were cleaned, recorded, and analyzed using STATA/SE version 14.0 statistical software. Sample weight was employed to manage sampling errors and non-responses. Continuous variables were categorized, and categorical variables were further re-categorized. Descriptive analysis was carried out to present the data in frequencies and percentages. Both the individual and community-level variables were presented using descriptive statistics. The DHS data's variables were organized in clusters; 71,503 under-five children are nested within households, and households were nested within 1690 clusters. The assumptions of independent observations and equal variance across clusters were broken to employ the traditional logistic regression model. This is an indication that using a sophisticated model to take into account between-cluster factors is necessary. As a result, multilevel mixed-effects logistic regression was used to determine the factors associated with prompt treatment of fever. Multilevel mixed effect logistic regression follows four models: the null model (outcome variable only), model I (only individual-level variables), model II (only community-level variables), and model III (both individual and community-level variables). The model without independent variables (the null model) was used to check the variability of prompt treatment of fever across the cluster. The association of individual-level variables with the outcome variable (Model I) and the association of community-level variables with the outcome variable (Model II) were assessed. In the final model (Model III), the association of both individual and community-level variables was fitted simultaneously with the outcome variable (prompt treatment of fever) [24].

The magnitude of the clustering effect and the degree to which community-level factors explain the unexplained variance of the null model were quantified by checking the intra-class correlation coefficient (ICC) and proportional change in variance (PCV). A model with the lowest deviance was selected as the best-fitted model. Finally, variables with a p-value less than 0.05 and an adjusted odds ratio (AOR) with a 95% confidence interval (CI) were described as statistically significant variables associated with prompt treatment of fever. The presence of multi-collinearity between covariates was checked by using a variance inflation factor (VIF) falling within acceptable limits of 1–10, indicating the absence of significant collinearity across independent variables.

## Random-effect results

Random effects or measures of variation of the outcome variable were estimated using the median odds ratio (MOR), ICC, and PCV. The variation between clusters was measured by the ICC and PCV. Taking clusters as a random variable, the ICC reveals that the variation of prompt treatment of fever between clusters is computed as ICC = $VC/(VC+3.29) \times 100\%$. The MOR is the median value of the odds ratio between the area of the highest risk and the area of the lowest risk for prompt treatment of fever when two clusters are randomly selected, using clusters as a random variable; MOR = $e^{0.95\sqrt{VC}}$. In addition, the PCV demonstrates the variation in the prevalence of prompt treatment of fever explained by factors and computed as; PCV = $(Vnull-VC)/Vnull \times 100\%$; where Vnull = variance of the null model and VC = cluster level variance [25]. The fixed effects were used to estimate the association between the likelihood of prompt treatment of fever and individual and community-level independent variables.

## Ethics statement

Permission was granted to download and use the data from http://www.dhs.program.com before conducting the study. Ethical clearance was obtained from the Institution Review Board of the DHS Program, ICF International. The procedures for DHS public-use data sets

were approved by the Institution Review Board. Identifiers for respondents, households, or sample communities were not allowed in any way, and the names of individuals or household addresses were not included in the data files. The number for each EA in the data file does not have labels to show their names or locations. There were no patients or members of the public involved since this study used a publicly available data set. As the study uses a retrospective study of archived samples, we have ensured that all data were fully anonymized before we accessed them from the IRB.

## Results

### Individual- and community-level characteristics of study subjects

The mean age of respondents was 28.90 ± 0.03 years, and 47.40% of them fall in the age range of 25–34 years. The majority (86.23%) of mothers were married, and 37.74% of them had no formal education. More than three-fourths (75.45%) of mothers had jobs, and 61.90% of them had media exposure. More than half (50.25%) of mothers had a poor household wealth index, and 56.56% of them reported that distance to a health facility was not a big problem. About 40.16% of mothers reported that healthcare decisions were made by husbands or partners alone, and 70.67% of them visited healthcare facilities in the last 12 months. More than two-thirds (67.38%) of mothers had four or more ANC visits during pregnancy, and 70.24% of them wanted their pregnancy. About 68.60% of mothers gave birth at a health facility. The mean age of children was 27.08 ± 0.08 months, and 53.36% of them fall in the age range of 24–59 months. More than half (50.88%) of children were male, and the majority (85.72%) of them had a long birth interval. Only 27.34% of mothers were from urban areas, and 51.85% of them were from a community with low media exposure. More than half (55.08%) and 53.35% of mothers were from a community with low education and high poverty levels, respectively. About 50.35% of mothers were from communities with high ANC utilization, and 53.82% of them were from communities with high health facility delivery (Table 2).

### Pooled prevalence of prompt treatment of fever

In the present study, the pooled prevalence of prompt treatment of fever among under-five children in sub-Saharan African countries was 26.11% (95% CI: 25.79%, 26.44%) (Fig 1). The result showed that the proportion of children who get prompt treatment for fever increases with increasing household wealth status, with the proportion of children who get prompt treatment for fever being lowest among children from poor household families (24.43%) and highest among rich household families (28.70%) (Fig 2). Similarly, the proportion of prompt treatment for fever was highest among mothers with secondary or higher education (29.68%) and lowest among mothers with no formal education (20.85%) (Fig 3).

### Measures of variation and model fitness

A null model was used to determine whether the data supported the decision to assess randomness at the community level. Findings from the null model showed that there were significant differences in prompt treatment of fever between communities, with a variance of 0.1909212 and a P value of <0.001. The variance within clusters contributed 94.51% of the variation in prompt treatment of fever, while the variance across clusters was responsible for 5.49% of the variation. In the null model, the odds of prompt treatment of fever differed between higher- and lower-risk clusters by a factor of 1.52 times. The intra-class correlation value for Model I indicated that 1.89% of the variation in prompt treatment of fever accounts for the disparities between communities. Then, with the null model, we used community-level

**Table 2. Individual-and community-level characteristics of study subjects, pooled data from 36 SSA countries, DHS 2006–2022.**

| Variables | Category | Frequency (n) | Percentage (%) |
|---|---|---|---|
| Mother's age | 15–24 years | 21,237 | 29.70 |
| | 25–34 years | 33,889 | 47.40 |
| | 35–49 years | 16,377 | 22.90 |
| Mother's education | No formal education | 26,986 | 37.74 |
| | Primary | 25,592 | 35.79 |
| | Secondary/higher | 18,925 | 26.47 |
| Mother's current marital status | Unmarried | 9,666 | 13.77 |
| | Married | 60,507 | 86.23 |
| Mother's working status | Not working | 17,552 | 24.55 |
| | Working | 53,951 | 75.45 |
| Media exposure | No | 27,194 | 38.10 |
| | Yes | 44,189 | 61.90 |
| Household wealth index | Poor | 35,931 | 50.25 |
| | Middle | 13,896 | 19.43 |
| | Rich | 21,676 | 30.32 |
| Distance to a health facility | Big problem | 28,224 | 43.44 |
| | Not a big problem | 36,750 | 56.56 |
| Health care decisions | Mothers alone | 10,252 | 14.34 |
| | With husband/partner | 21,039 | 29.43 |
| | Husband/partner alone | 28,719 | 40.16 |
| | Others | 11,493 | 16.07 |
| Visited healthcare facility last 12 months | No | 20,133 | 29.33 |
| | Yes | 48,500 | 70.67 |
| Pregnancy intention | Unintended | 20,457 | 29.76 |
| | Intended | 48,294 | 70.24 |
| Frequency of antenatal care use | None | 5,116 | 7.15 |
| | 1–3 visits | 18,209 | 25.47 |
| | 4+ visits | 48,178 | 67.38 |
| Place of delivery | Home | 22,450 | 31.40 |
| | Health facility | 49,053 | 68.60 |
| Child age | 0–11 months | 9,955 | 21.39 |
| | 12–23 months | 11,750 | 25.25 |
| | 24–59 months | 24,831 | 53.36 |
| Child sex | Male | 36,380 | 50.88 |
| | Female | 35,123 | 49.12 |
| Preceding birth interval | <24 months | 10,208 | 14.28 |
| | ≥24 months | 61,295 | 85.72 |
| Place of residence | Urban | 19,550 | 27.34 |
| | Rural | 51,953 | 72.66 |
| Community-level media exposure | Low | 37,077 | 51.85 |
| | High | 34,426 | 48.15 |
| Community-level education | Low | 39,381 | 55.08 |
| | High | 32,122 | 44.92 |
| Community poverty level | Low | 33,353 | 46.65 |
| | High | 38,150 | 53.35 |
| Community-level of ANC utilization | Low | 35,504 | 49.65 |
| | High | 35,999 | 50.35 |
| Community-level health facility delivery | Low | 33,017 | 46.18 |
| | High | 38,486 | 53.82 |

variables to generate Model II. According to the ICC value from Model II, cluster variations were the basis for 5.11% of the differences in prompt treatment of fever. In the final model (model III), which attributed approximately 1.82% of the variation in the likelihood of prompt treatment of fever to both individual and community-level variables, the likelihood of prompt treatment of fever varied by 1.26 times across low and high prompt treatment of fever (Table 3).

### Individual and community-level factors associated with prompt treatment of fever

In the final fitted model of multivariable multilevel logistic regression, maternal educational level, maternal occupation, media exposure, household wealth index, distance to a health facility, healthcare decisions, visited healthcare facility last 12 months, ANC visits attended during pregnancy, place of delivery, and community-level ANC utilization were significantly associated with prompt treatment of fever among under-five children in sub-Saharan African countries.

The odds of prompt treatment of fever were 1.18 and 1.15 times higher among mothers with primary and secondary education compared with mothers with no formal education, respectively [AOR = 1.18; 95% CI (1.13, 1.25)] and [AOR = 1.15; 95% CI (1.08, 1.23)]. Mothers who had jobs were 1.34 times more likely to have prompt treatment for fever compared with their counterparts [AOR = 1.34; 95% CI (1.27, 1.41)]. Mothers who listen to radio, watch

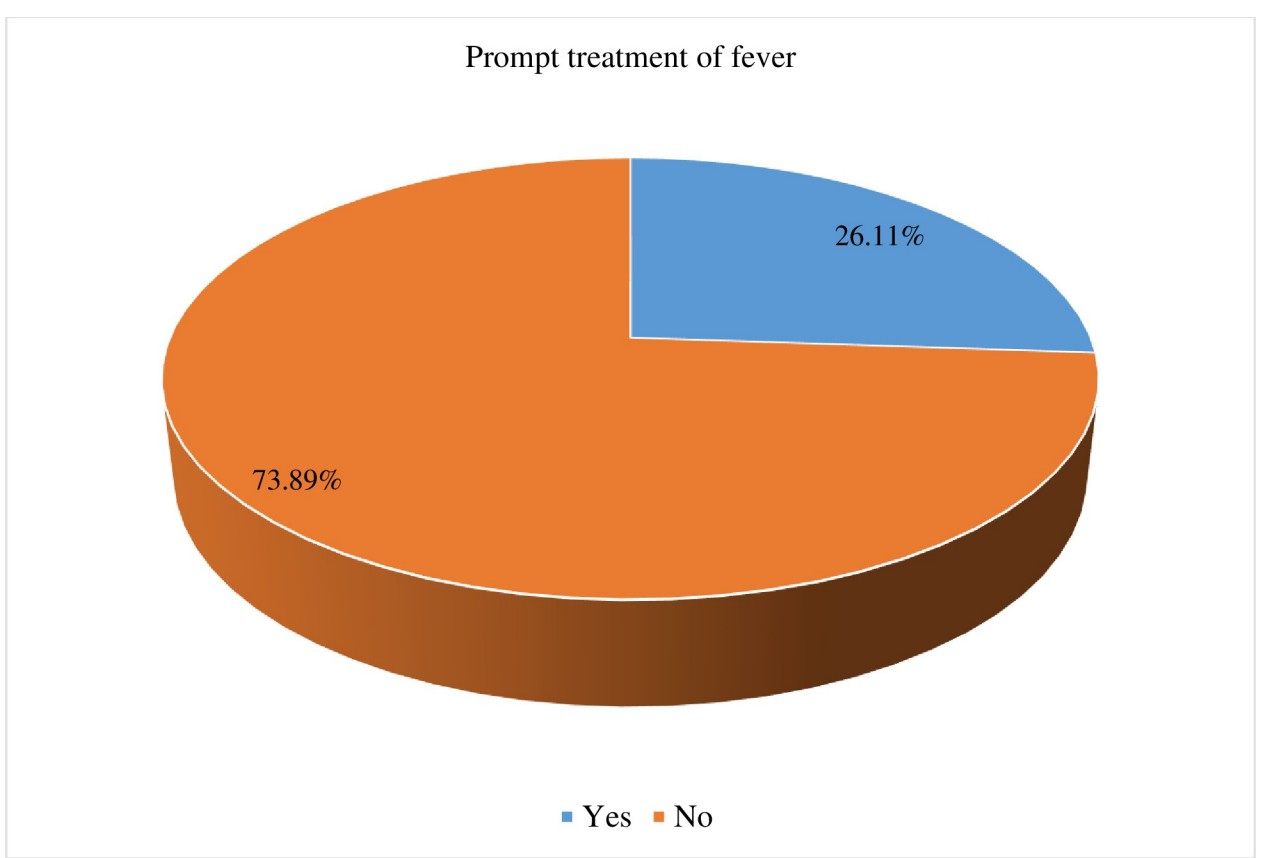

**Fig 1. Prevalence of prompt treatment of fever among under-five children in sub-Saharan African countries, DHS 2006–2022 (n = 71,503).**

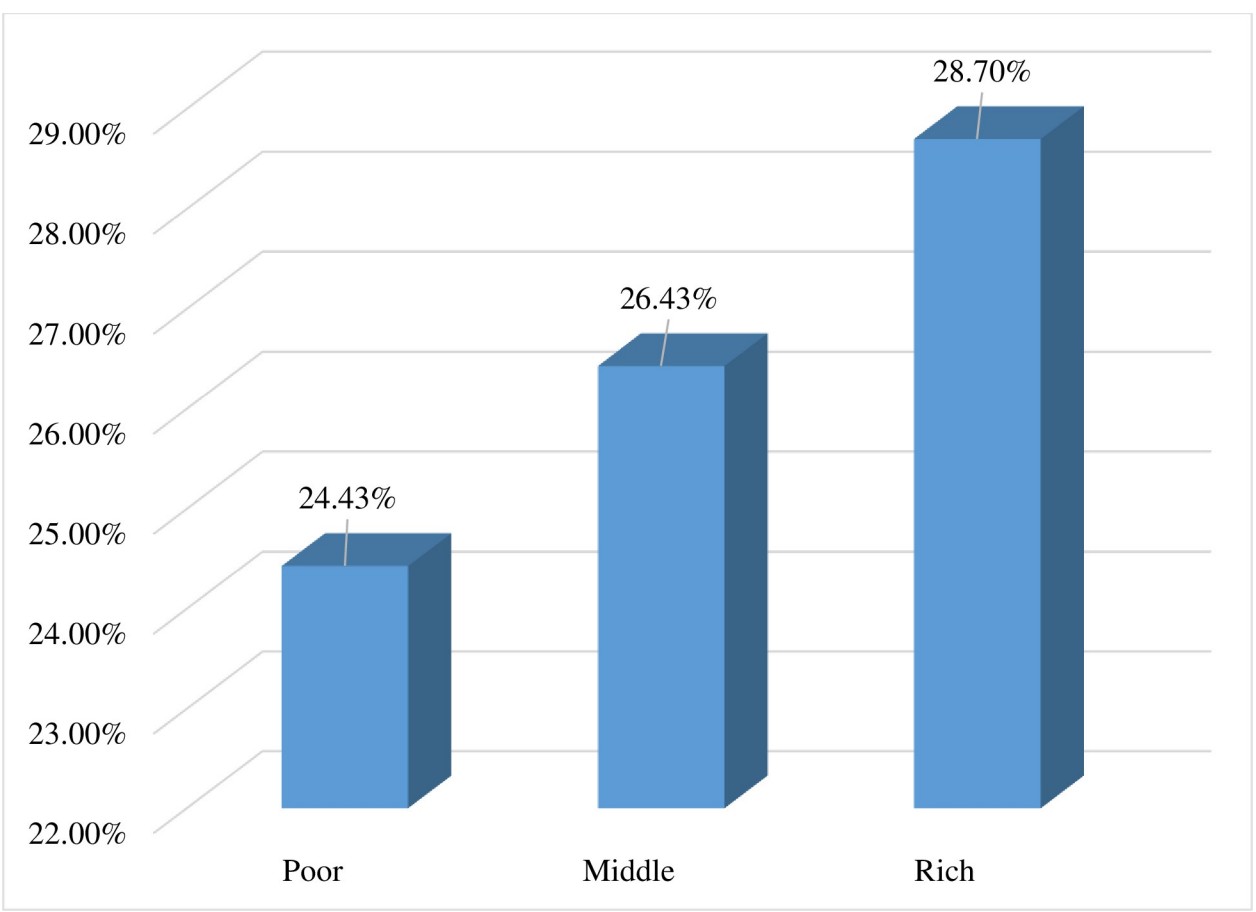

**Fig 2. Proportion of prompt treatment of fever by household wealth status among under-five children in sub-Saharan African countries, DHS 2006–2022 (n = 71,503).**

television, and read newspapers were 1.05 times more likely to have prompt treatment for fever than those who had no media exposure [AOR = 1.05; 95% CI (1.01, 1.10)]. Children of mothers with rich household wealth status were 1.13 times more likely to have prompt treatment of fever compared with those from poor household wealth status [AOR = 1.13; 95% CI (1.06, 1.19)]. Mothers who reported distance to a health facility was not a big problem were 1.18 times more likely to have prompt treatment of fever than their counterparts [AOR = 1.18; 95% CI (1.13, 1.23)]. The odds of prompt treatment of fever were 1.34 times higher among mothers who made healthcare decisions with their husband or partner than those who made decisions with someone else [AOR = 1.34; 95% CI (1.01, 1.77)].

Similarly, mothers who visited healthcare facilities in the last 12 months were 1.45 times more likely to have prompt treatment for fever compared with those who didn't [AOR = 1.45; 95% CI (1.38, 1.52)]. The frequency of ANC use was another factor significantly associated with prompt treatment of fever among under-five children. Accordingly, mothers who attended 1–3 and 4+ ANC visits were 1.57 and 1.79 times more likely to have prompt treatment of fever compared with mothers who didn't attend ANC visits during pregnancy, respectively [AOR = 1.57; 95% CI (1.41, 1.74)] and [AOR = 1.79; 95% CI (1.61, 1.99)]. The odds of prompt treatment of fever were 1.55 times higher among mothers who gave birth at a health facility compared with those who gave birth at home [AOR = 1.55; 95% CI (1.47, 1.63)]. From

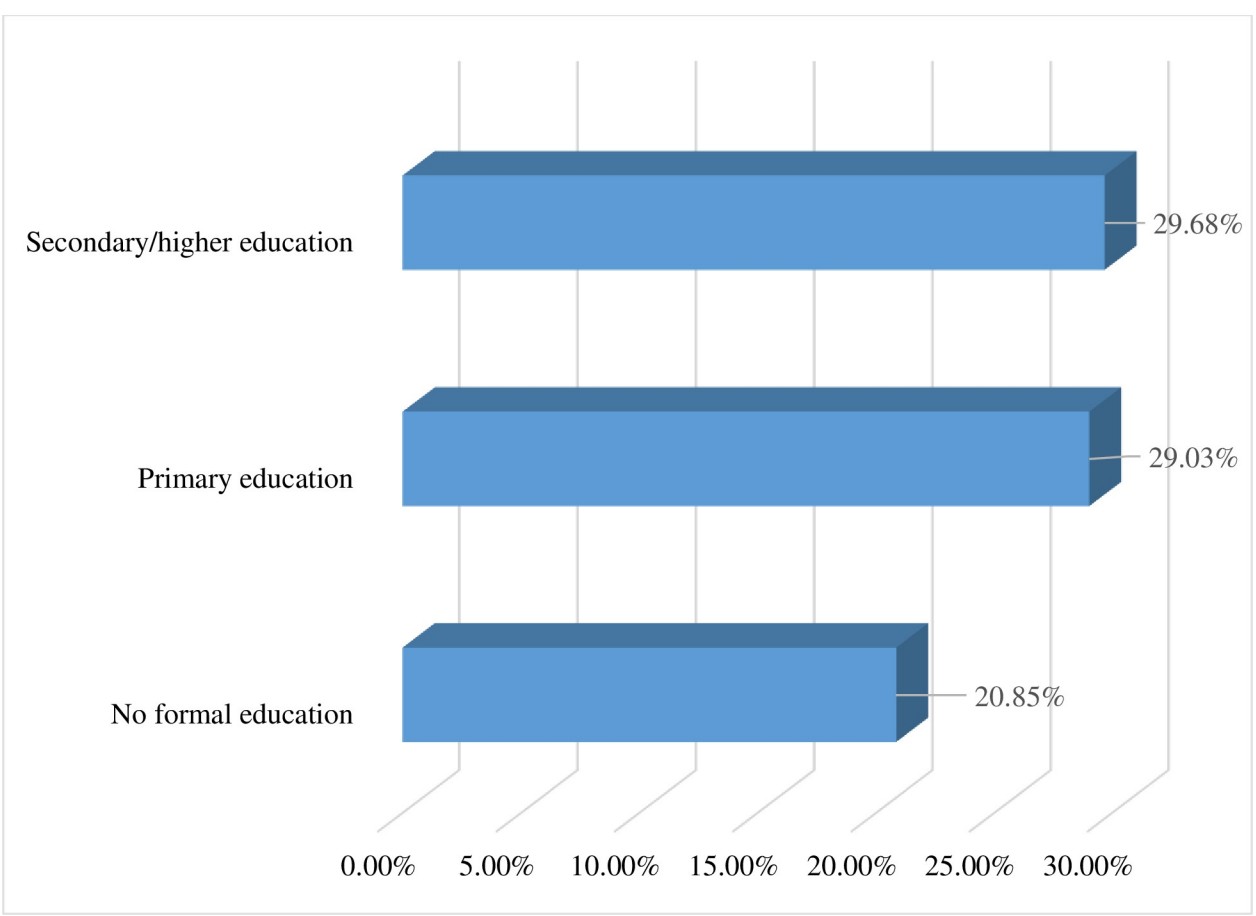

**Fig 3. Proportion of prompt treatment of fever by educational status among under-five children in sub-Saharan African countries, DHS 2006–2022 (n = 71,503).**

community-level variables, community-level ANC utilization was significantly associated with prompt treatment of fever among under-five children. Children of mothers from a community with high community-level ANC utilization were 1.08 times more likely to have prompt treatment of fever compared with those from a community with low community-level ANC utilization [AOR = 1.08; 95% CI (1.02, 1.14)] (Table 4).

**Table 3. Model comparison and random effect analysis for prompt treatment of fever and its associated factors in SSA countries, DHS 2006–2022 (n = 71,503).**

| Parameter | Null model | Model I | Model II | Model III |
|---|---|---|---|---|
| Variance | 0.1909212 | 0.0632378 | 0.1772304 | 0.061124 |
| ICC | 5.49% | 1.89% | 5.11% | 1.82% |
| MOR | 1.52 | 1.27 | 1.49 | 1.26 |
| PCV | Reference | 66.88% | 7.17% | 67.98% |
| **Model fitness** | | | | |
| LLR | -40731.897 | -27248.049 | -40663.021 | -27244.342 |
| Deviance | 81463.794 | 54496.098 | 81326.042 | 54488.684 |

ICC: Intra cluster correlation; LLR: log-likelihood ratio; MOR: median odds ratio; PCV: Proportional change in variance

**Table 4. Multivariable multilevel logistic regression analysis of individual and community-level factors associated with prompt treatment of fever among under-five children in SSA countries.**

| Variables | Category | Model I AOR (95% CI) | Model II AOR (95% CI) | Model III AOR (95% CI) |
|---|---|---|---|---|
| Mother's age | 15–24 years | 1 | | 1 |
| | 25–34 years | 1.01(0.96,1.05) | | 1.01(0.95,1.05) |
| | 35–49 years | 0.99(0.94,1.06) | | 1.00(0.94,1.06) |
| Mother's education | No formal education | 1 | | 1 |
| | Primary | 1.19(1.13,1.25)* | | 1.18(1.13,1.25)* |
| | Secondary/higher | 1.16(1.09,1.23)* | | 1.15(1.08,1.23)* |
| Current marital status | Unmarried | 1 | | 1 |
| | Married | 0.79(0.60,1.05) | | 0.79(0.60,1.05) |
| Mother's working status | Not working | 1 | | 1 |
| | Working | 1.34(1.27,1.41)* | | 1.34(1.27,1.41)* |
| Media exposure | No | 1 | | 1 |
| | Yes | 1.05(1.00,1.10)* | | 1.05(1.01,1.10)* |
| Household wealth index | Poor | 1 | | 1 |
| | Middle | 1.01(0.95,1.06) | | 1.00(0.95,1.06) |
| | Rich | 1.13(1.07,1.19)* | | 1.13(1.06,1.19)* |
| Distance to a health facility | Big problem | 1 | | 1 |
| | Not a big problem | 1.18 (1.13,1.23)* | | 1.18(1.13,1.23)* |
| Healthcare decisions | Mothers alone | 1.20(0.90,1.58) | | 1.19(0.90,1.58) |
| | With partner | 1.34(1.01,1.77)* | | 1.34(1.01,1.77)* |
| | Partner alone | 1.22(0.92,1.61) | | 1.22(0.92,1.61) |
| | Others | 1 | | 1 |
| Visited health facility | No | 1 | | 1 |
| | Yes | 1.45(1.38,1.52)* | | 1.45(1.38,1.52)* |
| Pregnancy intention | Unintended | 1 | | 1 |
| | Intended | 1.02(0.98,1.07) | | 1.02(0.98,1.07) |
| Frequency of ANC use | None | 1 | | 1 |
| | 1–3 visits | 1.57(1.41,1.75)* | | 1.57(1.41,1.74)* |
| | 4+ visits | 1.80(1.62,2.00)* | | 1.79(1.61,1.99)* |
| Place of delivery | Home | 1 | | 1 |
| | Health facility | 1.55(1.47,1.63)* | | 1.55(1.47,1.63)* |
| Child age | 0–11 months | 1 | | 1 |
| | 12–23 months | 1.03(0.98,1.10) | | 1.03(0.98,1.10) |
| | 24–59 months | 1.04(0.99,1.10) | | 1.04(0.99,1.10) |
| Child sex | Male | 1.01(0.97,1.05) | | 1.01(0.97,1.05) |
| | Female | 1 | | 1 |
| Preceding birth interval | <24 months | 1 | | 1 |
| | ≥24 months | 1.02(0.96,1.08) | | 1.02(0.96,1.08) |
| Place of residence | Urban | | 1.11(1.07,1.16)* | 1.01(0.95,1.06) |
| | Rural | | 1 | 1 |
| Community media exposure | Low | | 1 | 1 |
| | High | | 1.12(1.04,1.21)* | 0.98(0.92,1.04) |
| Community-level education | Low | | 1 | 1 |
| | High | | 1.30(1.21,1.40)* | 1.01(0.95,1.07) |
| Community poverty level | Low | | 1.02(0.95,1.10) | 0.99(0.93,1.05) |
| | High | | 1 | 1 |
| Community ANC utilization | Low | | 1 | 1 |
| | High | | 1.07(1.00,1.15)* | 1.08(1.02,1.14)* |
| Community-level health facility delivery | Low | | 1 | 1 |
| | High | | 0.94 (0.88,1.01) | 0.97(0.91,1.03) |

## Discussion

As far as the researcher's knowledge is concerned, this is the first study to determine the pooled prevalence and associated factors of prompt treatment of fever among under-five children in SSA using nationally representative DHS data. In the current study, the pooled prevalence of prompt treatment of fever among under-five children in SSA countries was 26.11% (95% CI: 25.79%, 26.44%). This finding was lower than studies conducted in Nigeria (37.9%) [15], Tanzania (44.6%) [14], Mazabuka District, Zambia (42%) [18], Zambia (57.3%) [19], and India (37.9%) [13]. The possible justification for this discrepancy might be due to differences in study setting, geographical variation, sample size, sociocultural differences, and differences in health service accessibility between countries. The difference might also be due to differences in the perceptions of caregivers of under-five children towards fever and febrile illnesses.

The study also identified individual and community-level variables associated with prompt treatment for fever. Accordingly, the odds of prompt treatment of fever were higher among educated mothers compared with mothers with no formal education. This finding was consistent with a study conducted in Zambia [19]. This might be due to the fact that education plays an important role in mothers' treatment-seeking behavior for their febrile children. Studies also indicated that there is a positive association between mothers' education and their health care-seeking behavior [26, 27]. This might also be explained by the fact that educated women are more likely to know the importance of seeking timely treatment for children during illnesses. Mothers who had jobs were more likely to have prompt treatment for fever compared with their counterparts. This finding was in agreement with a study conducted in Nigeria [15]. This might be due to the fact that working mothers have the capability to pay for management whenever they present their febrile children to a health facility for treatment. This could also be attributed to working mothers having a higher income, being highly educated, and having adequate resources to seek prompt treatment for their febrile children, which enables them to visit healthcare facilities for fever before it gets worse. Mothers who had media exposure were more likely to have prompt treatment for fever. This finding was supported by a study conducted in Woldia, Ethiopia [28], Chad [29], and India [27]. This might be due to mothers' access to radio, television, and newspapers increasing their understanding of the importance of child healthcare, and this may describe their higher likelihood of prompt treatment of fever. Media exposure might also be associated with rich wealth status, which can also facilitate timely treatment seeking for febrile children. Media exposure is an essential instrument for health promotion and has a substantial positive influence on healthcare service utilization [30].

Children of mothers with rich household wealth status were more likely to have prompt treatment for fever compared with those with poor household wealth status. Studies conducted in Nigeria [15] and India [13] reported similar findings. This might be due to the fact that mothers with poor wealth status may not be able to afford the cost of healthcare services, and this is possible to prevent them from seeking prompt treatment for their febrile children. Mothers who reported that distance to a health facility was not a big problem were more likely to have prompt treatment for fever than their counterparts. This finding was supported by studies conducted in Tanzania [14] and India [13]. This might be due to the fact that the challenges faced by mothers in accessing healthcare facilities decreased when they reported that distance to a health facility was not a big problem. A short distance to healthcare facilities may also decrease transportation costs to and from the facility, which may further motivate mothers to seek early treatment for fever. The odds of prompt treatment of fever were higher among mothers who made healthcare decisions with their husband or partner than among those who made decisions with someone else. This finding was consistent with a study conducted in Nigeria [31]. This might be explained by the influence of men over household resources and

decision-making to simplify treatment seeking and traverse challenges in accessing care for children [32].

Likewise, mothers who visited healthcare facilities in the last 12 months were more likely to receive prompt treatment for fever compared with those who didn't. This might be due to frequent visits to healthcare facilities, which could increase the chances of recommending timely care-seeking behavior and increase awareness of the importance of seeking timely and appropriate medical care for febrile illnesses. Antenatal care attendance was another factor significantly associated with prompt treatment of fever, in which mothers who attended ANC visits and had high community-level ANC utilization were more likely to have prompt treatment of fever compared with their counterparts. This finding was consistent with studies conducted in Nigeria [15] and Burkina Faso [33]. This might be due to the fact that antenatal care attendance gives mothers the opportunity to acquire information on maternal and child health promotion [34]. Accessing ANC services in LMICs is directly associated with improved birth outcomes and longer-term decreases in child mortality [35]. Furthermore, the odds of prompt treatment for fever were higher among mothers who gave birth at a health facility compared with those who gave birth at home. This finding was in agreement with studies conducted in Ethiopia [36] and Nigeria [37]. This might be due to the fact that mothers who gave birth at a health facility might have better awareness and trust about healthcare services, easily understand the significance of seeking healthcare services, and understand the availability of services wanted for the treatment of their febrile children. Giving birth at a health facility increases the interest of mothers in healthcare services, including healthcare professionals, and inspires them to strive for treatment when their child shows febrile disease manifestations [38].

## Implications of the study

The findings of this study could be used as an input for policymakers and governmental and non-governmental organizations, specifically those who work on child health, to design proper intervention strategies both at national and regional levels, as pooled national survey data is used.

## Strengths and limitations of the study

This study has the following strengths: Firstly, the study used weighted nationally representative data from 36 sub-Saharan African countries with a large sample size. Secondly, to accommodate the hierarchical nature of the DHS data and get a reliable standard error and estimate, a multilevel analysis was employed. This study also had some limitations. There might be a possibility of recall and social desirability bias as the DHS survey was based on respondents' self-reports. The cross-section nature of the data couldn't enable us to show the temporal relationship between variables. Besides, we couldn't incorporate variables like knowledge and perceptions of mothers towards the treatment of fever due to the secondary nature of the data.

## Conclusion

The pooled prevalence of prompt treatment of fever among under-five children in sub-Saharan African countries was low. Educated women, working mothers, having media exposure, rich household wealth status, perceiving distance to a health facility was not a big problem, making healthcare decisions with husband or partner, visiting healthcare facility in the last 12 months, ANC attendance, health facility delivery, and high community-level ANC utilization increase the odds of prompt treatment of fever. Therefore, women empowerment, information dissemination through mass media, maintaining regular visits to healthcare facilities, strengthening health facility delivery and antenatal care services, and giving prior attention to non-

working mothers and those from poor household wealth status are strongly recommended to improve early treatment seeking behavior among mothers of under-five febrile children.

## Acknowledgments

We are grateful to the DHS program for letting us use the relevant DHS data in this study.

## Author Contributions

**Conceptualization:** Enyew Getaneh Mekonen, Belayneh Shetie Workneh, Alebachew Ferede Zegeye.

**Data curation:** Enyew Getaneh Mekonen, Belayneh Shetie Workneh, Alebachew Ferede Zegeye.

**Formal analysis:** Enyew Getaneh Mekonen, Belayneh Shetie Workneh, Alebachew Ferede Zegeye.

**Investigation:** Enyew Getaneh Mekonen, Belayneh Shetie Workneh, Alebachew Ferede Zegeye.

**Methodology:** Enyew Getaneh Mekonen, Belayneh Shetie Workneh, Alebachew Ferede Zegeye.

**Software:** Enyew Getaneh Mekonen, Belayneh Shetie Workneh, Alebachew Ferede Zegeye.

**Supervision:** Enyew Getaneh Mekonen, Tadesse Tarik Tamir, Alebachew Ferede Zegeye.

**Validation:** Enyew Getaneh Mekonen, Tadesse Tarik Tamir, Alebachew Ferede Zegeye.

**Visualization:** Enyew Getaneh Mekonen, Belayneh Shetie Workneh, Tadesse Tarik Tamir.

**Writing – original draft:** Enyew Getaneh Mekonen, Belayneh Shetie Workneh, Tadesse Tarik Tamir.

**Writing – review & editing:** Enyew Getaneh Mekonen, Belayneh Shetie Workneh, Tadesse Tarik Tamir, Alebachew Ferede Zegeye.

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
