## [Decision Letter · Decision Letter 0]

12 Mar 2024

PONE-D-24-00421Prompt Treatment of Fever and its Associated Factors among Under-five Children in sub-Saharan Africa: A Multilevel Analysis of Evidence from 36 CountriesPLOS ONE

Dear Dr. Mekonen,

Thank you for submitting your manuscript to PLOS ONE. After careful consideration, we feel that it has merit but does not fully meet PLOS ONE’s publication criteria as it currently stands. Therefore, we invite you to submit a revised version of the manuscript that addresses the points raised during the review process.

We look forward to receiving your revised manuscript.

Kind regards,

Kahsu Gebrekidan

Academic Editor

PLOS ONE

Journal Requirements:

Whilst you may use any professional scientific editing service of your choice, PLOS has partnered with both American Journal Experts (AJE) and Editage to provide discounted services to PLOS authors. Both organizations have experience helping authors meet PLOS guidelines and can provide language editing, translation, manuscript formatting, and figure formatting to ensure your manuscript meets our submission guidelines. To take advantage of our partnership with AJE, visit the AJE website (http://aje.com/go/plos) for a 15% discount off AJE services. To take advantage of our partnership with Editage, visit the Editage website (www.editage.com) and enter referral code PLOSEDIT for a 15% discount off Editage services. If the PLOS editorial team finds any language issues in text that either AJE or Editage has edited, the service provider will re-edit the text for free.

https://pubmed.ncbi.nlm.nih.gov/38033152/

file:///home/jasmine/Downloads/journal.pone.0262411.pdf

In your revision ensure you cite all your sources (including your own works), and quote or rephrase any duplicated text outside the methods section. Further consideration is dependent on these concerns being addressed.

4. In the online submission form, you indicated that Data available on request from the author.

Reviewers' comments:

Reviewer's Responses to Questions

**Comments to the Author**

1. Is the manuscript technically sound, and do the data support the conclusions?

Reviewer #1: No

Reviewer #2: Yes

2. Has the statistical analysis been performed appropriately and rigorously? 

Reviewer #1: Yes

Reviewer #2: Yes

3. Have the authors made all data underlying the findings in their manuscript fully available?

Reviewer #1: Yes

Reviewer #2: Yes

4. Is the manuscript presented in an intelligible fashion and written in standard English?

Reviewer #1: Yes

Reviewer #2: Yes

5. Review Comments to the Author

Reviewer #1: no

Reviewer #2: I suggest that the figures' labels in the manuscript's body be placed directly alongside the figures, similar to how tables are labeled. This will enhance the manuscript's clarity and readability, facilitating better understanding for readers. Additionally, I suggest adding the study's implications, which would make the manuscript more sound.

6. PLOS authors have the option to publish the peer review history of their article (what does this mean?). If published, this will include your full peer review and any attached files.

Reviewer #1: No

Reviewer #2: No

---

## [Author Response · Author response to Decision Letter 0]

15 Mar 2024

Response to Reviewers and Editors

Editor comments 1: Please ensure that your manuscript meets PLOS ONE's style requirements, including those for file naming.

Author’s response: Considered.

Editor comments 2: We suggest you thoroughly copyedit your manuscript for language usage, spelling, and grammar. If you do not know anyone who can help you do this, you may wish to consider employing a professional scientific editing service.

Author’s response: We have addressed it accordingly.

Editor comments 3: We noticed you have some minor occurrence of overlapping text with the following previous publication(s), which needs to be addressed.

Author’s response: We have addressed it accordingly.

Editor comments 4: In the online submission form, you indicated that Data available on request from the author.

Author’s response: We used DHS data available online at https://dhsprogram.com/data/available-datasets.cfm.

Editor comments 5: Please review your reference list to ensure that it is complete and correct.

Author’s response: We have revised it as per the journal guideline.

Reviewer 2: I suggest that the figures' labels in the manuscript's body be placed directly alongside the figures, similar to how tables are labeled. This will enhance the manuscript's clarity and readability, facilitating better understanding for readers. Additionally, I suggest adding the study's implications, which would make the manuscript more sound.

Author’s response: Thank you very much and we have considered it.

---

## [Decision Letter · Decision Letter 1]

30 Apr 2024

Prompt Treatment of Fever and its Associated Factors among Under-five Children in sub-Saharan Africa: A Multilevel Analysis of Evidence from 36 Countries

PONE-D-24-00421R1

Dear Mr. Enyew,

We’re pleased to inform you that your manuscript has been judged scientifically suitable for publication and will be formally accepted for publication once it meets all outstanding technical requirements.

Kind regards,

Kahsu Gebrekidan

Academic Editor

PLOS ONE

Additional Editor Comments (optional):

Reviewers' comments:

Reviewer's Responses to Questions

**Comments to the Author**

1. If the authors have adequately addressed your comments raised in a previous round of review and you feel that this manuscript is now acceptable for publication, you may indicate that here to bypass the “Comments to the Author” section, enter your conflict of interest statement in the “Confidential to Editor” section, and submit your "Accept" recommendation.

Reviewer #2: All comments have been addressed

2. Is the manuscript technically sound, and do the data support the conclusions?

Reviewer #2: Yes

3. Has the statistical analysis been performed appropriately and rigorously? 

Reviewer #2: Yes

4. Have the authors made all data underlying the findings in their manuscript fully available?

Reviewer #2: Yes

5. Is the manuscript presented in an intelligible fashion and written in standard English?

Reviewer #2: Yes

6. Review Comments to the Author

Reviewer #2: (No Response)

7. PLOS authors have the option to publish the peer review history of their article (what does this mean?). If published, this will include your full peer review and any attached files.

Reviewer #2: No

---

## [Editor Report · Acceptance letter]

3 May 2024

PONE-D-24-00421R1 

PLOS ONE

Dear Dr. Mekonen, 

I'm pleased to inform you that your manuscript has been deemed suitable for publication in PLOS ONE. Congratulations! Your manuscript is now being handed over to our production team.

Kind regards, 

on behalf of

Dr. Kahsu Gebrekidan 

Academic Editor

PLOS ONE